# Association between sites and severity of eczema and the onset of cow's milk and egg allergy in children

Shiori Kawada[1¤], Masaki Futamura[1,2]*, Hiroya Hashimoto[2], Manabu Ono[1], Nobuhiro Akita[1], Masahiro Sekimizu[1,2], Hiroyoshi Hattori[1,2], Masahiko Goto[1], Keizo Horibe[1], Naoko Maeda[1]

1 Department of Pediatrics, National Hospital Organization Nagoya Medical Center, Nagoya, Japan,
2 Clinical Research Center, National Hospital Organization Nagoya Medical Center, Nagoya, Japan

¤ Current address: Department of Pediatrics, Nakatsugawa Municipal General Hospital, Gifu, Japan
* masakifutamura@gmail.com

**Data Availability Statement:** All relevant data are within the manuscript and Supporting Information files.

## Abstract

### Background

Cow's milk allergy (CMA) and egg allergy (EA) are common and can reduce quality of life in children. Infantile eczema is a well-established risk factor for the onset of food allergy via transdermal sensitization; however, various types of infantile eczema have not yet been evaluated. Therefore, we assessed the association between CMA and EA and the sites and the severity of infantile eczema.

### Methods

This retrospective study was based on data from patients aged 2–19 years with atopic disease who were treated between July 2015 and March 2019 in a pediatric allergy clinic in Japan. Data regarding the history of IgE-mediated symptoms, eczema in the first year of life, parental history of atopic diseases, and infantile nutrition were collected.

### Results

A total of 289 patients were included in the study, of which 81 and 111 children had IgE-mediated CMA and EA, respectively. The rates of CMA and EA were higher in the children with infantile eczema than in those without (30% vs. 9% and 42% vs. 21%). The rate of CMA was also higher in children with eczema on the face. Significant differences were noted in the rate of CMA among children with facial eczema of exudation (adjusted odds ratio 2.398; P = 0.017) and papules (adjusted odds ratio 2.787; P = 0.008), using multivariate analysis.

### Conclusion

The rate of IgE-mediated CMA was high among children with atopic disease having severe facial eczema during infancy.

**Funding:** Financial assistance was provided by Maruho Co., Ltd. The funder had no role in study design, data collection and analysis, decision to publish, or preparation of the manuscript.

**Competing interests:** MF has received consultancy, lecture fees, and honoraria for lecturing from Maruho Co., Ltd. The remaining authors have nothing to disclose. This does not alter our adherence to PLOS ONE policies on sharing data and materials.

## Introduction

Food allergy is a frequently occurring disease in children in numerous countries [1]. Cow's milk and egg are the most frequently reported antigens involved in food allergy, and the incidence of immunoglobulin E (IgE)-mediated cow's milk allergy (CMA) and egg allergy (EA) are 2.3% and 2.5% in Europe, respectively [2]. Provided that cow's milk is the primary source of infantile nutrition following breast milk, the presence of CMA greatly impairs the quality of life of children and their caregivers [3]. Egg is also a key nutrient which provides fatty acids, vitamins and proteins and benefit to child nutrition and brain development [4].

The reported incidence of food allergy ranges from 6% to 10% in Europe, the United States, and Japan [5–7]. The incidence of food allergy has increased to 8% over the past two decades, according to a survey conducted in 98 countries worldwide in the third phase of the International Study of Asthma and Allergies in Childhood [1]. Transdermal sensitization is a factor extensively involved in the development of food allergy [8], and interventions aimed at preventing eczema onset are expected to prevent transdermal sensitization [9, 10]. The establishment of effective prevention methods can limit the development of subsequent allergic diseases which is termed as "allergic march" [11].

Notably, early-onset and severe eczema in infancy were reported to be strongly associated with food allergy [8, 12]. However, to the best of our knowledge, no studies have evaluated the relationship of infantile eczema based on the sites involved in or specific types of food allergy. Therefore, the present study aimed to investigate the characteristics of infantile eczema in the first year of life that were associated with CMA onset among atopic children. Additionally, we also investigated the association between infantile eczema and EA.

## Materials and methods

This study was retrospective in nature. We collected clinical data of pediatric patients 1) who had single or multiple atopic diseases; 2) who were aged 2–19 years; and 3) who visited the Pediatric Allergy Clinic in National Hospital Organization Nagoya Medical Center, Japan, between July 2015 and March 2019. According to the institutional review board, patients meeting these criteria were enrolled in the study unless their caregivers opted out of data sharing. Patients and caregivers did not provide informed written consent, but they could access detailed information of this study on the website. The study data were collected from electronic medical records with patient identifiers and included infantile eczema, previous immediate allergic symptoms to cow's milk or egg, history of other atopic diseases, parental history of atopic diseases, parental history of smoking during pregnancy, ingestion of cow milk or its products during breastfeeding period and the first year of life, and age at solid food introduction.

The cow's milk-specific and egg white-specific serum IgE antibody levels were measured using ImmunoCAP® test (Phadia AB, Uppsala, Sweden) to assess sensitization. In the current study, CMA was defined as positive for cow's milk-specific IgE antibody ≥0.35 IU/ml accompanied with a history of immediate allergic symptoms after the intake of cow's milk or its products. EA was also defined as immediate allergic symptom after intake of egg products with positive egg white-specific IgE antibody. We did not include any non-IgE-mediated symptoms to cow's milk or egg.

Caregivers of all patients were asked whether their children presented with eczema in the first year of life and, if so, to indicate the affected sites including the head, face, trunk, and upper and lower limbs. In particular, data of the presence of symptoms including redness, dryness, exudation and/or papules on the face were collected for children with facial eczema.

The study analyses included a comparison of the differences in skin conditions and other factors between children with and without each specific food allergy. Pearson's chi-squared test was used for between-group comparisons and a logistic regression analysis was used to calculate odds ratios. Spearman's correlation coefficient was calculated for correlation between different sites of infantile eczema. The multiplicity of tests was adjusted by Holm-Bonferroni correction. P-values of less than 0.05 was considered statistically significant. All statistical analyses were performed using SAS® statistical software version 9.4 (SAS Institute, Cary, NC, USA).

The study protocol was approved by the institutional review board of the National Hospital Organization Nagoya Medical Center (reference number 2019–008).

## Results

A total of 289 children aged 2–19 years were included in the present study. In the study cohort, there were 81 (28%) children with CMA and 111(38%) with EA, and infantile eczema was confirmed in 84% of the children. The demographic data of the study cohort are shown in Table 1. No factor related to infantile nutrition, including sex, parental atopic disease history and parental smoking during pregnancy, were significantly associated with CMA and EA. Although there was no relationship between cow's milk intake in infancy and CMA ($P$ = 1.000), the proportion of children with cow's milk intake after 7 months of age was significantly lower in patients with CMA than in those without CMA (11% vs. 37%, $P$ < 0.001). The incidence of CMA was significantly higher in the group with longer breastfeeding periods. However, there was no statistical relationship with cow's milk intake or breastfeeding period in patients with EA. More than 80% of the children were introduced to solid food between 5 and 7 months of age. None of the children with EA were introduced to solid food before four months of age. There was no significant high prevalence of CMA and EA among the age groups of solid food introduction.

More than 80% of the children had episodes of infantile eczema. Positive correlations were observed between all eczema site pairs. Strong correlation was found between the upper and lower limbs (correlation coefficient r = 0.83). Moderate correlations were found between the trunk and the upper limb (r = 0.69) and between the trunk and the lower limb (r = 0.65). The other pairs were weakly correlated (all r < 0.40).

CMA and EA incidence were higher in children with infantile eczema than in those without infantile eczema (30% vs. 9% and 42% vs. 21%, respectively) (Table 2). In univariate analysis of the affected sites, CMA incidence was significantly higher in children with eczema on the face than in those without these. However, the factor in CMA and EA was not significant in multivariate analysis (adjusted odd ratio 3.496, P = 0.078).The analysis of the characteristics of face eczema lesions revealed that the incidence of CMA was significantly higher in children with erythema than in those without erythema (33% vs. 20%, $P$ = 0.043); in children with exudation than in those without exudation (40% vs. 22%, $P$ = 0.011); and in those with papules than in those without papules (41% vs. 23%, $P$ = 0.017). These differences in the proportion of CMA based on the presence of exudation and papules remained significant in the multivariate analysis (adjusted odd ratio, 2.398 and 2.787; $P$ = 0.017 and $P$ = 0.008, respectively) (Table 3). In the multivariate analysis, children with any face lesions did not have significantly different proportion of EA as compared to those without it.

## Discussion

The current retrospective study revealed that the presence of severe facial eczema in children with atopic disease was associated with the development of CMA, although there was no association between CMA and eczema at other sites.

**Table 1. Characteristics of the study population.**

| Characteristics | | n | cow's milk allergy | | egg allergy | |
|---|---|---|---|---|---|---|
| | | | prevalence (%) | P-value | prevalence (%) | P-value |
| Sex | | | | | | |
| | Female | 118 | 28 | 1.000 | 34 | 0.219 |
| | Male | 171 | 28 | | 42 | |
| Parental atopic disease | | | | | | |
| | Yes | 252 | 30 | 0.218 | 40 | 0.185 |
| | No | 33 | 18 | | 27 | |
| Atopic dermatitis | | | | | | |
| | Yes | 134 | 27 | 0.696 | 43 | 0.185 |
| | No | 155 | 29 | | 35 | |
| Bronchial asthma | | | | | | |
| | Yes | 72 | 19 | 0.070 | 33 | 0.331 |
| | No | 217 | 31 | | 40 | |
| Smoking during pregnancy | | | | | | |
| | Yes | 93 | 28 | 1.000 | 38 | 0.896 |
| | No | 181 | 28 | | 39 | |
| Cow's milk intake from 0 to 12 months of age | | | | | | |
| | Yes | 174 | 27 | 1.000 | 41 | 0.886 |
| | No | 72 | 26 | | 39 | |
| Cow's milk intake from 0 to 6 months of age | | | | | | |
| | Yes | 167 | 28 | 0.879 | 41 | 0.783 |
| | No | 80 | 26 | | 39 | |
| Cow's milk intake from 7 to 12 months of age | | | | | | |
| | Yes | 85 | 11 | **<0.001** | 35 | 0.273 |
| | No | 152 | 37 | | 43 | |
| Breastfeeding duration in infancy | | | | | | |
| | 0 to 5 | 42 | 10 | **0.011** | 29 | 0.204 |
| | 6 to 11 | 38 | 29 | | 47 | |
| | 12 and more | 179 | 31 | | 41 | |
| Age of solid food introduction | | | | | | |
| | Before 4 months | 8 | 13 | 0.598 | 0 | - |
| | 5–7 months | 216 | 29 | | 42 | |
| | ≥8 months | 33 | 24 | | 33 | |

The immature skin barrier at this age might underlie the relationship between infantile eczema and food allergy, because an immature skin barrier has been shown to enhance transdermal allergic sensitization and to contribute to the onset of food allergy [13].

In the current study, the proportion of CMA and EA was higher in children with infantile eczema. The maturity of skin barriers is determined by number of horny cells and transepidermal water loss (TEWL); higher TEWL is observed in infancy [14]. Conversely, the TEWL declines and the stratum corneum becomes thicker until the age of four years [14]; small and immature keratinocytes are associated with higher TEWL.

We also found that CMA was significantly more frequent in children with infantile eczema on the face. Next to the skin in the genital areas, face has the smallest horny cells and is one of the locations with highest TEWL in the entire body. In addition, the serine proteases kallikrein 5 and kallikrein 7, which are most abundant in cheek keratinocytes [15], are highly expressed

**Table 2. Association between food allergy and the site of infantile eczema.**

| Site | | n | cow's milk allergy | | | | | | | egg allergy | | | | | | |
|---|---|---|---|---|---|---|---|---|---|---|---|---|---|---|---|---|
| | | | prevalence (%) | OR | 95%CI | P-value | aOR† | 95%CI | P-value | prevalence (%) | OR | 95%CI | P-value | aOR† | 95%CI | P-value |
| Any | Y | 242 | 30 | 4.464 | 1.322–15.065 | 0.080 | 4.726 | 1.074–20.789 | 0.200 | 42 | 2.763 | 1.158–6.593 | 0.110 | 1.995 | 0.800–4.975 | 0.693 |
| | N | 34 | 9 | | | | | | | 21 | | | | | | |
| Head | Y | 120 | 33 | 1.652 | 0.959–2.845 | 0.282 | 1.495 | 0.833–2.684 | 0.711 | 43 | 1.205 | 0.734–1.978 | 1.000 | 0.99 | 0.583–1.678 | 1.000 |
| | N | 142 | 23 | | | | | | | 38 | | | | | | |
| Face | Y | 206 | 33 | **4.017** | **1.640–9.836** | **0.014** | 3.496 | 1.302–9.388 | 0.078 | 44 | 2.374 | 1.222–4.613 | 0.065 | 1.881 | 0.922–3.838 | 0.496 |
| | N | 56 | 11 | | | | | | | 25 | | | | | | |
| Trunk | Y | 154 | 29 | 1.180 | 0.678–2.051 | 0.862 | 1.132 | 0.622–2.061 | 1.000 | 41 | 1.088 | 0.658–1.799 | 1.000 | 1.024 | 0.599–1.751 | 1.000 |
| | N | 108 | 26 | | | | | | | 39 | | | | | | |
| Upper limbs | Y | 155 | 30 | 1.250 | 0.717–2.180 | 0.862 | 1.006 | 0.554–1.827 | 1.000 | 40 | 0.992 | 0.600–1.640 | 1.000 | 0.977 | 0.571–1.671 | 1.000 |
| | N | 107 | 25 | | | | | | | 40 | | | | | | |
| Lower limbs | Y | 154 | 31 | 1.385 | 0.792–2.422 | 0.759 | 1.124 | 0.615–2.054 | 1.000 | 38 | 0.837 | 0.507–1.381 | 1.000 | 0.738 | 0.430–1.266 | 1.000 |
| | N | 108 | 24 | | | | | | | 43 | | | | | | |

Y, yes; N, no; OR, odds ratio; aOR, adjusted odds ratio; CI, confidence interval

P- values were adjusted by Holm- Bonferroni correction.

†adjusted by gender, parental atopic disease, and cow's milk intake during first six months of life

in epidermal keratinocytes; these proteases promote the desquamation of the horny layer, thereby lowering the skin barrier. These factors might underlie much promotion of CMA by severe eczema with exudation or papules.

**Table 3. Association between food allergy and the lesion of facial eczema.**

| Eczema | | n | cow's milk allergy | | | | | | | egg allergy | | | | | | |
|---|---|---|---|---|---|---|---|---|---|---|---|---|---|---|---|---|
| | | | prevalence (%) | OR | 95%CI | P-value | aOR† | 95%CI | P-value | prevalence (%) | OR | 95%CI | P-value | aOR† | 95%CI | P-value |
| Erythema | Y | 155 | 33 | **1.985** | **1.108–3.557** | **0.043** | 1.647 | 0.880–3.082 | 0.237 | 46 | 1.919 | 1.143–3.222 | 0.055 | 1.751 | 1.007–3.044 | 0.189 |
| | N | 106 | 20 | | | | | | | 31 | | | | | | |
| Xerosis | Y | 93 | 30 | 1.214 | 0.693–2.127 | 0.498 | 1.137 | 0.625–2.068 | 0.675 | 44 | 1.281 | 0.766–2.143 | 1.000 | 1.226 | 0.716–2.100 | 1.000 |
| | N | 168 | 26 | | | | | | | 38 | | | | | | |
| Exudation | Y | 86 | 40 | **2.357** | **1.344–4.136** | **0.011** | **2.398** | **1.290–4.455** | **0.017** | 42 | 1.106 | 0.654–1.869 | 1.000 | 1.056 | 0.600–1.860 | 1.000 |
| | N | 175 | 22 | | | | | | | 39 | | | | | | |
| Papule | Y | 66 | 41 | **2.308** | **1.275–4.176** | **0.017** | **2.787** | **1.458–5.329** | **0.008** | 42 | 1.129 | 0.641–1.989 | 1.000 | 0.992 | 0.540–1.822 | 1.000 |
| | N | 195 | 23 | | | | | | | 40 | | | | | | |

Y, yes; N, no; OR, odds ratio; aOR, adjusted odds ratio; CI, confidence interval

P- values were adjusted by Holm- Bonferroni correction.

†adjusted by gender, parental atopic disease, and cow's milk intake during first six months of life

Cow's milk and egg are the most common antigen involved in IgE-mediated food allergy in Japan [7]. Early intake of egg white and peanuts is recommended to prevent the development of IgE-mediated allergy [16]. In addition, the introduction of milk between four and six months of age was reported to prevent the onset of CMA [17]. Delaying the introduction of solid foods is not recommended for any children, even those with high allergy risk [18]. Most children in the present study had already been introduced to solid foods during the recommended periods before the first visit. Prior to the introduction of these recommendations, many caregivers had already started introducing the milk proteins before introducing the solid foods, which allowed the infants to have the opportunity to have contact with cow's milk on their skin in early life. Severe eczema on the face associated with damaged keratinocytes might have enhanced the sensitisation to cow's milk during infancy due to contact with the lesions. The results from previous studies suggest that earlier and more severe eczema in infancy might be due to more frequent food allergy [8, 12].

The current study showed that the rate of CMA was significantly lower in the children who were initiated on cow's milk intake after 7 months of age; however, the rate of EA was not significantly different in these children. Children with CMA were expected to have fewer opportunities for cow's milk intake, suggesting that most of the children developed CMA within 6 months of age because there was no correlation with cow's milk intake in the first year of life. The presence of CMA might have also prolonged the breastfeeding period; therefore, the initiation of cow's milk intake up to six months of age was included as a covariate in multivariate analysis.

The current study has several limitations. First, the retrospective study design did not allow us to determine the potential causal relationship between severe facial eczema and food allergy. We asked caregivers regarding the presence of eczema in the first year of life at the first visit when their children were 2 years or more of age. Therefore, most children were considered to have developed CMA after infantile eczema. Second, we did not include the age at introduction of egg as a covariate because this information was not included in the electronic medical records. Although most children in the current study had started solid food after 5 months of age, few children are introduced to egg products in early infancy. We cannot deny that the EA results could have been different with direct information about egg intake. However, we believe that it did not have a major effect on the CMA results. Third, we did not analyze details regarding the duration of breastfeeding or cow's milk intake. The results revealed that the proportion of CMA was not dependent on the duration of breastfeeding or the cow's milk intake at the age of six months, suggesting that the exclusive intake of breast milk, while not preventative against CMA, might prevent the overall development of atopic diseases. Finally, the current study cohort was limited to atopic children. It remains unclear whether the study findings can be expanded to the general population. However, we could observe specific relationship of eczema with CMA, but not with EA, in atopic children.

## Conclusions

In conclusion, the present retrospective study revealed that the incidence of IgE-mediated CMA was higher in atopic children with severe facial eczema during the first year of life. The cause-effect relationship between severe facial eczema and CMA and the relationship with other food allergies require evidence from future birth cohort studies.

## Supporting information

**S1 Dataset.**
(XLSX)

## Acknowledgments

We thank Ms. Mitsue Takemura for her contribution in extracting data from the electronic medical records. We also thank Enago (www.enago.jp) for the English language review.

## Author Contributions

**Conceptualization:** Masaki Futamura.

**Data curation:** Shiori Kawada, Masaki Futamura, Hiroya Hashimoto.

**Formal analysis:** Masaki Futamura, Hiroya Hashimoto.

**Funding acquisition:** Masaki Futamura.

**Investigation:** Masaki Futamura.

**Methodology:** Masaki Futamura.

**Project administration:** Masaki Futamura.

**Supervision:** Naoko Maeda.

**Validation:** Shiori Kawada.

**Writing – original draft:** Shiori Kawada, Masaki Futamura.

**Writing – review & editing:** Shiori Kawada, Masaki Futamura, Hiroya Hashimoto, Manabu Ono, Nobuhiro Akita, Masahiro Sekimizu, Hiroyoshi Hattori, Masahiko Goto, Keizo Horibe, Naoko Maeda.

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
