## [Decision Letter · Decision Letter 0]

16 Jul 2020

PONE-D-20-18145

Association between sites and severity of eczema and the onset of cow’s milk and egg allergy in children

PLOS ONE

Dear Dr. Futamura,

Thank you for submitting your manuscript to PLOS ONE. After careful consideration, we feel that it has merit but does not fully meet PLOS ONE’s publication criteria as it currently stands. Therefore, we invite you to submit a revised version of the manuscript that addresses the points raised during the review process.

We look forward to receiving your revised manuscript.

Kind regards,

Linglin Xie

Academic Editor

PLOS ONE

Journal Requirements:

2. In ethics statement in the manuscript and in the online submission form, please provide additional information about the patient records used in your retrospective study. Specifically, please ensure that you have discussed whether all data were fully anonymized before you accessed them and/or whether the IRB or ethics committee waived the requirement for informed consent. If patients provided informed written consent to have data from their medical records used in research, please include this information.

3.

We note that you have indicated that data from this study are available upon request. PLOS only allows data to be available upon request if there are legal or ethical restrictions on sharing data publicly. For information on unacceptable data access restrictions, please see http://journals.plos.org/plosone/s/data-availability#loc-unacceptable-data-access-restrictions.

Reviewers' comments:

Reviewer's Responses to Questions

**Comments to the Author**

1. Is the manuscript technically sound, and do the data support the conclusions?

Reviewer #1: Yes

Reviewer #2: Partly

2. Has the statistical analysis been performed appropriately and rigorously? 

Reviewer #1: Yes

Reviewer #2: Yes

3. Have the authors made all data underlying the findings in their manuscript fully available?

Reviewer #1: Yes

Reviewer #2: Yes

4. Is the manuscript presented in an intelligible fashion and written in standard English?

Reviewer #1: Yes

Reviewer #2: Yes

5. Review Comments to the Author

Reviewer #1: Major comments:

This article is a retrospective study on the association between CMA and EA and the sites and severity of infantile eczema. The study analyzed the data from patients aged 2 – 19 years with atopic disease and was treated in a pediatric allergy clinic. The data analysis has several limitations based on their experimental design, but the authors have clearly stated these limitations in their discussion. The data analyses support their conclusions that IgE-mediated CMA was positively correlated with severe facial eczema during infancy among children with atopic disease.

One major issue with the statistical analyses is that the author conducted multiple tests during their analyses. At the same time, they used a threshold P-value of 0.05 for significance, a typical cut-off when doing a single test. I would suggest that they use an adjusted p-value instead to control the false positive rate for this multiple test case.

Minor comments:

Line 204: It is not very clear why the limitation would allow the authors to observe the relationship they mentioned. Maybe the authors could make the expression clearer for the readers.

Reviewer #2: This manuscript describes a retrospective study about the relationship between eczema and cow milk or egg allergy in children. It demonstrated that the incidence of infantile eczema is higher in subjects with cow milk allergy, especially facial eczema. However, major concerns are also noticed as listed:

1. Language editing is recommended. Such as the usage of “whereas” in the Discussion section

2. Does the paper consider any potential correlation between different sites of infantile eczema? The paper mentioned 289 subjects are involved in the study, while according to Table 2, there seems to be a few subjects with multiple sites of infantile eczema.

3. In the Conclusion section, the paper states that the incidence of IgE-mediated CMA is higher in atopic children during their first year. However, in the “Material and methods” section, it says the collected data were from 2-19 years old patients. Meanwhile, I could not find any data mentioning any significant difference of CMA incidence during their first year.

6. PLOS authors have the option to publish the peer review history of their article (what does this mean?). If published, this will include your full peer review and any attached files.

Reviewer #1: No

Reviewer #2: **Yes: **Yushu Qin

---

## [Author Response · Author response to Decision Letter 0]

14 Sep 2020

Reviewer #1: Major comments:

This article is a retrospective study on the association between CMA and EA and the sites and severity of infantile eczema. The study analyzed the data from patients aged 2 – 19 years with atopic disease and was treated in a pediatric allergy clinic. The data analysis has several limitations based on their experimental design, but the authors have clearly stated these limitations in their discussion. The data analyses support their conclusions that IgE-mediated CMA was positively correlated with severe facial eczema during infancy among children with atopic disease.

One major issue with the statistical analyses is that the author conducted multiple tests during their analyses. At the same time, they used a threshold P-value of 0.05 for significance, a typical cut-off when doing a single test. I would suggest that they use an adjusted p-value instead to control the false positive rate for this multiple test case.

RESPONSE: We appreciate your suggestion. We have adjusted the p-values in Table 2 and 3 by Holm-Bonferroni correction and have added the following sentence in the Materials and Methods section; “The multiplicity of tests was adjusted by Holm-Bonferroni correction.” We have also changed the text accordingly. 

Minor comments:

Line 204: It is not very clear why the limitation would allow the authors to observe the relationship they mentioned. Maybe the authors could make the expression clearer for the readers.

RESPONSE: We agree that the meaning of that sentence was not clear for readers. We have changed it to “However, we could observe specific relationship of eczema with CMA, but not with EA, in atopic children.”

 

Reviewer #2: This manuscript describes a retrospective study about the relationship between eczema and cow milk or egg allergy in children. It demonstrated that the incidence of infantile eczema is higher in subjects with cow milk allergy, especially facial eczema. However, major concerns are also noticed as listed:

1. Language editing is recommended. Such as the usage of “whereas” in the Discussion section.

RESPONSE: We appreciate your recommendation. We have asked the language review to the company again and attach the certificate of English editing.

2. Does the paper consider any potential correlation between different sites of infantile eczema? The paper mentioned 289 subjects are involved in the study, while according to Table 2, there seems to be a few subjects with multiple sites of infantile eczema.

RESPONSE: We have analyzed the correlation between different sites of eczema. The results are presented below:

Spearman’s correlation coefficient of each site of infantile eczema (see the attached file)

We have added the following sentences in the Materials and Methods and in the Results; “Spearman’s correlation coefficient was calculated for correlation between different sites of infantile eczema” and “Positive correlations were observed between all eczema site pairs. Strong correlations were found between the upper and lower limbs (correlation coefficient r = 0.83) and moderate correlations were found between the trunk and the upper limb (r = 0.69) and between the trunk and the lower limb (r = 0.65). The other pairs were weakly correlated (all r < 0.40).”

3. In the Conclusion section, the paper states that the incidence of IgE-mediated CMA is higher in atopic children during their first year. However, in the “Material and methods” section, it says the collected data were from 2-19 years old patients. Meanwhile, I could not find any data mentioning any significant difference of CMA incidence during their first year.

RESPONSE: We have mentioned “Caregivers of all patients were asked whether their children presented with eczema in the first year of life” in the Materials and Methods. However, for clearer expression, we have changed the prior sentence to “The study data were collected from electronic medical records and included infantile eczema, previous immediate allergic symptoms…”

---

## [Decision Letter · Decision Letter 1]

7 Oct 2020

Association between sites and severity of eczema and the onset of cow’s milk and egg allergy in children

PONE-D-20-18145R1

Dear Dr. Futamura,

We’re pleased to inform you that your manuscript has been judged scientifically suitable for publication and will be formally accepted for publication once it meets all outstanding technical requirements.

Kind regards,

Linglin Xie

Academic Editor

PLOS ONE

Additional Editor Comments (optional):

Reviewers' comments:

Reviewer's Responses to Questions

**Comments to the Author**

1. If the authors have adequately addressed your comments raised in a previous round of review and you feel that this manuscript is now acceptable for publication, you may indicate that here to bypass the “Comments to the Author” section, enter your conflict of interest statement in the “Confidential to Editor” section, and submit your "Accept" recommendation.

Reviewer #1: All comments have been addressed

Reviewer #2: All comments have been addressed

2. Is the manuscript technically sound, and do the data support the conclusions?

Reviewer #1: Yes

Reviewer #2: Yes

3. Has the statistical analysis been performed appropriately and rigorously? 

Reviewer #1: Yes

Reviewer #2: (No Response)

4. Have the authors made all data underlying the findings in their manuscript fully available?

Reviewer #1: Yes

Reviewer #2: (No Response)

5. Is the manuscript presented in an intelligible fashion and written in standard English?

Reviewer #1: Yes

Reviewer #2: Yes

6. Review Comments to the Author

Reviewer #1: (No Response)

Reviewer #2: (No Response)

7. PLOS authors have the option to publish the peer review history of their article (what does this mean?). If published, this will include your full peer review and any attached files.

Reviewer #1: No

Reviewer #2: No

---

## [Editor Report · Acceptance letter]

9 Oct 2020

PONE-D-20-18145R1 

Association between sites and severity of eczema and the onset of cow’s milk and egg allergy in children 

Dear Dr. Futamura:

I'm pleased to inform you that your manuscript has been deemed suitable for publication in PLOS ONE. Congratulations! Your manuscript is now with our production department. 

Kind regards, 

on behalf of

Dr. Linglin Xie 

Academic Editor

PLOS ONE